# Exploring the Spatial Characteristics of Typhoon-Induced Vegetation Damages in the Southeast Coastal Area of China from 2000 to 2018

**Lizhen Lu [1,*]** , **Chuyi Wu [1]** and **Liping Di [2]**

1   School of Earth Sciences, Zhejiang University, 38 Zheda Rd, Hangzhou 310027, China; 3150102540@zju.edu.cn
2   Center for Spatial Information Science and Systems, George Mason University, 4400 University Drive, Fairfax, VA 22030, USA; ldi@gmu.edu
*   Correspondence: llz_gis@zju.edu.cn; Tel.: +86-571-8795-1336

**Abstract:** The southeast coastal area of China (SCAC), a typhoon-prone area with a long coastline, suffers severe damage from typhoons almost every year. Exploring the spatial characteristics of historical typhoon-induced vegetation damage (VD) is crucial to predicting VD after severe typhoon landfalls and improving strategies for vegetation protection and restoration. Remote sensing is an efficient and feasible approach for measuring large-scale VD caused by natural disasters. This paper, by exploring the spatial distribution of VD of every severe landfalling typhoon with Google Earth Engine (GEE), aims to reveal the spatial characteristics of typhoon-induced VD in SCAC. Firstly, the values of disaster vegetation damage index (DVDI), difference in enhanced vegetation index (DEVI), and normalized difference vegetation index (DNDVI) for the 28 selected landing typhoons in SCAC were calculated and compared by using moderate resolution imaging spectroradiometer (MODIS) data in GEE. Secondly, every DVDI image was overlaid with land cover, elevation, relative aspect and typhoon path layers in ArcGIS. Thirdly, spatial characteristics of VD were revealed with the aid of spatial statistical analysis. The study found that: (1) DVDI is a more effective index for evaluating VD caused by typhoons. (2) The Pearl River Delta is the most severe VD region. The severe VD regions for four typhoon groups have significantly spatial correlation with typhoon-landing locations. (3) Forests are ranked the first in terms of damaged areas by typhoon in every year, followed by sparse forests. (4) Topography has no influence on VD by a single typhoon event, and relative aspect has no correlation with VD caused by typhoons in SCAC.

**Keywords:** disaster vegetation damage index (DVDI); Google Earth Engine (GEE); southeast coastal area of China (SCAC); spatial characteristics; MODIS

## 1. Introduction

The southeast coastal area of China (SCAC), with its long coastline and location in a typhoon-active area, suffers severe damages from typhoons almost every year. According to statistics [1], 43 typhoons landed in SCAC from 2000 to 2018 and resulted in extensive vegetation damage (VD) and enormous economic loss. For instance, Typhoon Mangkhu swept through SCAC in 2018, when 174,400 hectares of crops were affected, of which 3300 hectares were destroyed (see Figure 1), and the direct economic loss was 5.2 billion yuan [2]; in 2013, Typhoon Utor damaged 4,278,400 croplands and induced 16.23 billion yuan of direct economic loss [3], etc. Understanding the spatial characteristics of historically typhoon-induced VD is crucial to projecting VD after severe typhoon landfall and improving strategies for vegetation protection and restoration.

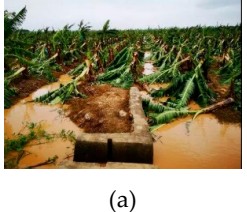 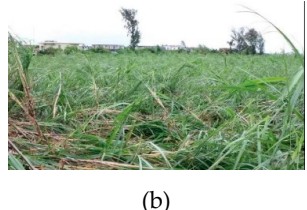 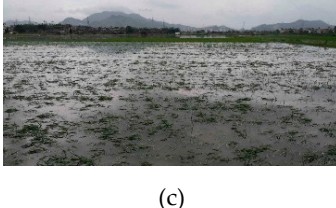

(a)                                    (b)                                    (c)

**Figure 1.** Pictures of vegetation damage (VD) induced by Mangkhu in 2018 (**a**) and (**b**) are from http://news.sina.com.cn/o/2018-09-19/doc-ihkhfqns8435454.shtml, (**c**) is from http://www.weather.com.cn/zt/tqzt/2930932.shtml.

Satellite remote sensing techniques are the primary efficient and feasible tools for extracting pixel-based information on vegetation damage induced by typhoons and other natural disasters at large scale. Many studies have evaluated VD successfully by calculating the differences in enhanced vegetation index (DEVI) or difference in normalized difference vegetation index (DNDVI) before and after a disaster via satellite remote sensing imagery [4–11]. For example, by using 250 m EVI imagery, the post hurricane forest damage of Hurricane Felix in 2007 was successfully characterized from pre- and postdisaster enhanced vegetation index (EVI) data [4]. Utilizing moderate resolution imaging spectroradiometer (MODIS, product MOD13Q1) time series imagery, pre- and post-disaster EVI values for five hurricanes' landfall along the northern Gulf of Mexico were derived, and the study results indicated that the MODIS data could be applied to detect severe hurricane damage to vegetation [5]. Using Landsat-8 OLI and Sentinel-2 data, VD on Dominica and Puerto Rico by Hurricane Maria (2017) was investigated by comparing the changes of NDVI in 2017 with those in reference years (2015 and 2016), with a sudden drop in NDVI values after Hurricane Maria's landfall detected [6]. The spatial pattern of damaged forest expressed as differences in NDVI (DNDVI) before and after Typhoon Saomai in 2006 were extracted from Landsat Enhanced Thematic Mapper Plus (ETM+) data, and the results showed that the influencing factors of DNDVI were vegetation aggregation, elevation, land use, relative aspect and distance from the typhoon's path [7]. Landsat-5 NDVI data before and after Hurricane Katrina's 2005 landfall in the WBR (Weeks Bay National Estuarine Research Reserve) indicated that the average NDVI values decreased by 49% after landfall [8].

The temporal differences of vegetation indices before and after a disastrous event, such as the aforementioned DNDVI and DEVI, have been used in various studies for characterizing the vegetative damage. Both NDVI and EVI are the measure of vegetative condition and health. The dynamic ranges of the indices for a specific location are related to vegetation types and their growth environment at the location. For example, at a sparsely vegetated location, the maximum NDVI value might be 0.2, and DNDVI of −0.2 represents the total destruction of vegetation at the location, while, at a densely vegetated location, the maximum NDVI value might be 0.8 and DNDVI of −0.2 represents slight damage to the vegetation. Therefore, DNDVI and DEVI cannot be used to compare the vegetative damage of different locations. To avoid this problem, Di et al. proposed the disaster vegetation damage index (DVDI) to calculate the difference of vegetation condition immediately before and after a natural disaster, and effectively measured the VD by flood disasters [12]. DVDI uses the difference of VCI, which has been normalized by location-specific historical vegetation dynamic ranges, making it comparable across locations. However, DVDI is a newly proposed index and has not been widely validated for different kind of natural disasters.

The aforementioned studies state that DVDI and the changes of EVI/NDVI pre- and post-disaster derived from satellite remote sensing imagery are effective indexes for evaluating VD due to typhoon events. For most existing studies, which focus on extracting VD induced by only one or a couple natural disasters, it is practical for computing VD indexes by downloading and compositing remote sensing data. But for studies investigating VD caused by many historical typhoons, the traditional computational approach for typhoon VD studies is not suitable due to time and labor-intensive data downloading, cost to acquire the data, and local computer power limitations. Applying Google

Earth Engine (GEE) can be more efficient due to its analysis-ready satellite image archive, scalable computing power, and integrated analysis capabilities. GEE is a cloud-based platform for processing very large geospatial datasets efficiently by avoiding scalability problems such as data acquisition and storage, parsing obscure file formats, central processing units (CPUs), graphics processing units (GPUs), managing databases, and machine allocations. GEE is also well designed for users and allows easy dissemination of results to others [13]. The GEE public data catalog is made up of earth-observing remote images, including the entire MODIS, Landsat, Sentinel-1&2 archives, as well as land cover data and many other geophysical, environmental and socio-economic datasets [13]. GEE is successfully and widely used in global forest change [14], flood mapping [15], crop yield estimation [16], land use change assessment [17], etc.

This study has two objectives: (1) to test whether DVDI can be applied to evaluate VD caused by typhoons in GEE; (2) to explore the spatial characteristics of VD by historical landing typhoons in SCAC.

## 2. Study Area and Data

### 2.1. The Study Area

The study area is in the southeast coastal area of China. It is centered at (23°37′, 110°46′), covers 417, 500 km$^2$, and includes Guangdong province, Guangxi Zhuang autonomous region, and the Hong Kong and Macao special administrative regions (see Figure 2). With a long coastline, the study area captures the overall trend of high northwest and low southeast. Its annual average precipitation is 1500–2214 mm (http://www.gd.gov.cn/, http://www.gxzf.gov.cn/). Its altitude ranges from 24 m to 2141.5 m, with an average elevation of 319.5 m. The Pearl River is the main water system in this area, and forms the Pearl River Delta, one of the most developed areas in China. The major crops in this area include rice, peanut, sugar cane, and plantain, among which sugar cane and plantain are vulnerable to strong wind disasters. Forests, sparse forests, croplands, and impervious lands covered about 45.2%, 35.8%, 12.3%, and 3.8% of the total area in 2016, respectively, as counted by using the MODIS land cover product (MCD12Q1).

### 2.2. Data

#### 2.2.1. The Selected Typhoons

The 43 typhoons that landed in SCAC between 2000 and 2018, recorded by the typhoon network of the China meteorological station, were checked, and the 28 of them whose landfall wind speed scale was equal or above 10 were selected. The information on these 28 typhoons is listed in Table 1. In order to analyze the spatial characteristics of vegetation damage induced by typhoon disasters, these 28 typhoons were sorted into four groups by their landfall locations: Leizhou Peninsula, Pearl River Delta, Eastern Guangzhou and Western Guangzhou. These four groups contain 3, 11, 7 and 7 typhoons respectively (see Figure 2 and Table 1). Figure 2 also shows that 22 of these 28 typhoons' paths are from southeast to northwest, indicating that the majority of landing typhoons affected almost the entire SCAC.

#### 2.2.2. Remote Sensing Data

The study area is located in the coastal region and frequently covered by clouds, so most of the time useful information cannot be extracted from a single remote sensing image alone. Because of frequent global coverage, MODIS allows cloud-free images for large geographic area during a short period to be obtained [18]. The GEE platform was applied to generate MODIS image sequences at 250 m spatial resolution before and after the selected typhoon events. Considering the short time of typhoon landing and extinction (see Table 1), and the effectiveness of VD assessment, the image sequences of 14 days before landing and 14 days after extinction, respectively, were used to reduce

composite images with lowest cloud composite at the pixel level in GEE. Practically, for each pixel, we checked from the closest landing/extinction date image by QC(quality check) 250 m band, found the first "good quality" value, and then composited it into the pre-/post-typhoon image.

**Table 1.** Information on the 28 selected typhoons.

| No. | ID | Name | Scale of Landfall Wind Speed | Time | Group | Correlation Coefficient (R) | Significance Level | Ratio (%) |
|---|---|---|---|---|---|---|---|---|
| 1 | 200013 | Maria | 10 | 2000.8.29–9.2 | Group 2 | 0.33 | 0.39 | 0.54 |
| 2 | 200104 | Utor | 11 | 2001.7.1–7.7 | Group 2 | −0.97 | 0.00 | 0.33 |
| 3 | 200107 | Yutu | 12 | 2001.7.22–7.26 | Group 4 | 0.77 | 0.01 | 0.77 |
| 4 | 200212 | Kammuri | 10 | 2002.8.1–8.6 | Group 3 | −0.92 | 0.00 | 2.87 |
| 5 | 200214 | Vongfong | 11 | 2002.8.15–8.20 | Group 4 | −0.97 | 0.00 | 0.00 |
| 6 | 200308 | Imbudo | 14 | 2003.7.15–7.25 | Group 4 | 0.90 | 0.00 | 0.88 |
| 7 | 200313 | Dujuan | 12 | 2003.8.28–9.3 | Group 2 | 0.14 | 0.69 | 0.5 |
| 8 | 200510 | Sanvu | 11 | 2005.8.9–8.15 | Group 3 | −0.80 | 0.01 | 2.49 |
| 9 | 200606 | Prapiroon | 12 | 2006.7.28–8.5 | Group 4 | 0.89 | 0.00 | 2.91 |
| 10 | 200812 | Nuri | 12 | 2008.8.17–8.23 | Group 2 | 0.92 | 0.00 | 0.31 |
| 11 | 200814 | Hagupit | 15 | 2008.9.17–9.25 | Group 4 | 0.91 | 0.00 | 1.10 |
| 12 | 200906 | Molave | 13 | 2009.7.15–7.19 | Group 2 | −0.59 | 0.09 | 0.00 |
| 13 | 200915 | Koppu | 12 | 2009.9.12–9.15 | Group 2 | 0.72 | 0.03 | 0.00 |
| 14 | 201003 | Chanthu | 12 | 2010.7.19–7.23 | Group 4 | 0.70 | 0.03 | 0.00 |
| 15 | 201011 | Fanapi | 12 | 2010.9.15–9.21 | Group 3 | −0.93 | 0.00 | 0.00 |
| 16 | 201208 | Vicente | 13 | 2012.7.21–7.25 | Group 2 | 0.79 | 0.01 | 0.00 |
| 17 | 201213 | Kai-tak | 12 | 2012.8.13–8.18 | Group 1 | 0.96 | 0.00 | 0.00 |
| 18 | 201311 | Utor | 14 | 2013.8.10–8.16 | Group 4 | −0.93 | 0.00 | 0.00 |
| 19 | 201319 | Usagi | 14 | 2013.9.19–9.23 | Group 3 | −0.97 | 0.00 | 0 |
| 20 | 201409 | Rammasun | 17 | 2014.7.12–7.20 | Group 1 | −0.15 | 0.70 | 0 |
| 21 | 201415 | Kalmaegi | 13 | 2014.9.12–9.17 | Group 1 | -0.14 | 0.72 | 0 |
| 22 | 201510 | Linfa | 12 | 2015.7.2–7.10 | Group 3 | 0.85 | 0.00 | 0 |
| 23 | 201522 | Mujigea | 15 | 2015.10.2–10.5 | Group 1 | −0.95 | 0.00 | 0 |
| 24 | 201604 | Nida | 14 | 2016.7.30–8.3 | Group 2 | −0.99 | 0.00 | 0.19 |
| 25 | 201622 | Haima | 14 | 2016.10.15–10.22 | Group 3 | 0.81 | 0.01 | 0 |
| 26 | 201713 | Hato | 14 | 2017.8.20–8.24 | Group 2 | 0.97 | 0.00 | 0 |
| 27 | 201714 | Pakhar | 12 | 2017.8.25–8.28 | Group 2 | 0.97 | 0.00 | 0 |
| 28 | 201822 | Mangkhut | 14 | 2018.9.7–9.17 | Group 2 | −0.94 | 0.00 | 0 |

Group 1: The Leizhou Peninsula landfalls group, three typhoons; Group 2: The Pearl River Delta landfalls group, 11 typhoons; Group 3: The Eastern Guangdong landfalls group, seven typhoons; Group 4: The Western Guangdong Landfalls group, seven typhoons. Correlation coefficient (*R*) and Significance level are the results of correlation analysis of elevation levels and area percentages of VD. Ratio is the percentage of pixels with bad quality using the images acquired within 7 days after the typhoon landfall to composite the post-typhoon image.

### 2.2.3. Land Cover Data

MCD12Q1 has an annual updating cycle and five classification schemes. From the aspect of compatibility degree between each classification scheme and land cover types in our study area, MCD12Q1 land cover type 2 of the university of Maryland (UMD) scheme at 500 m resolution was used as our original data and reclassified them into nine types: (referring to [19]) croplands, forests, grasslands, shrublands, wetlands, water bodies, impervious land, barelands, and sparse forest (see Table 2). Due to the lack of MCD12Q1 2000, MCD12Q1 2001 was used for 2000, assuming that land cover change in adjacent years can be ignored for statistics. The four dominant types of land cover, i.e., forests, sparse forests, croplands, and impervious lands covered 97.11% of total area. The other five land cover types covered less than 3% in 2016 in SCAC.

### 2.2.4. DEM

The Advanced Spaceborne Thermal Emission Reflection Radiometer (ASTER) Generalized Digital Environment Model (GDEM) data at 30 m spatial resolution were downloaded from the Geospatial data cloud website (http://www.gscloud.cn/). These GDEM data are used for deriving relative aspects to typhoon paths, and also for exploring the relationship between VDs and elevations.

**Table 2.** Land cover types in this study and the moderate resolution imaging spectroradiometer (MODIS) University of Maryland (UMD) scheme.

| Land Cover Types in this Study | Percentages of Area in SCAC in 2016 | MODIS Land Cover Types of UMD Scheme |
|---|---|---|
| Forests | 45.23% | Evergreen Needleleaf and Broadleaf Forests, Deciduous Broadleaf Forests, Mixed Forests |
| Sparse Forests | 35.83% | Woody Savannas |
| Croplands | 12.27% | Croplands, Cropland/Natural Vegetation Mosaics |
| Impervious Lands | 3.78% | Urban and Built-up Lands |
| Grasslands | 1.22% | Grasslands |
| Wetlands | 0.81% | Permanent Wetlands |
| Water Bodies | 0.80% | Water Bodies |
| Barelands | 0.06% | Non-Vegetated Lands |

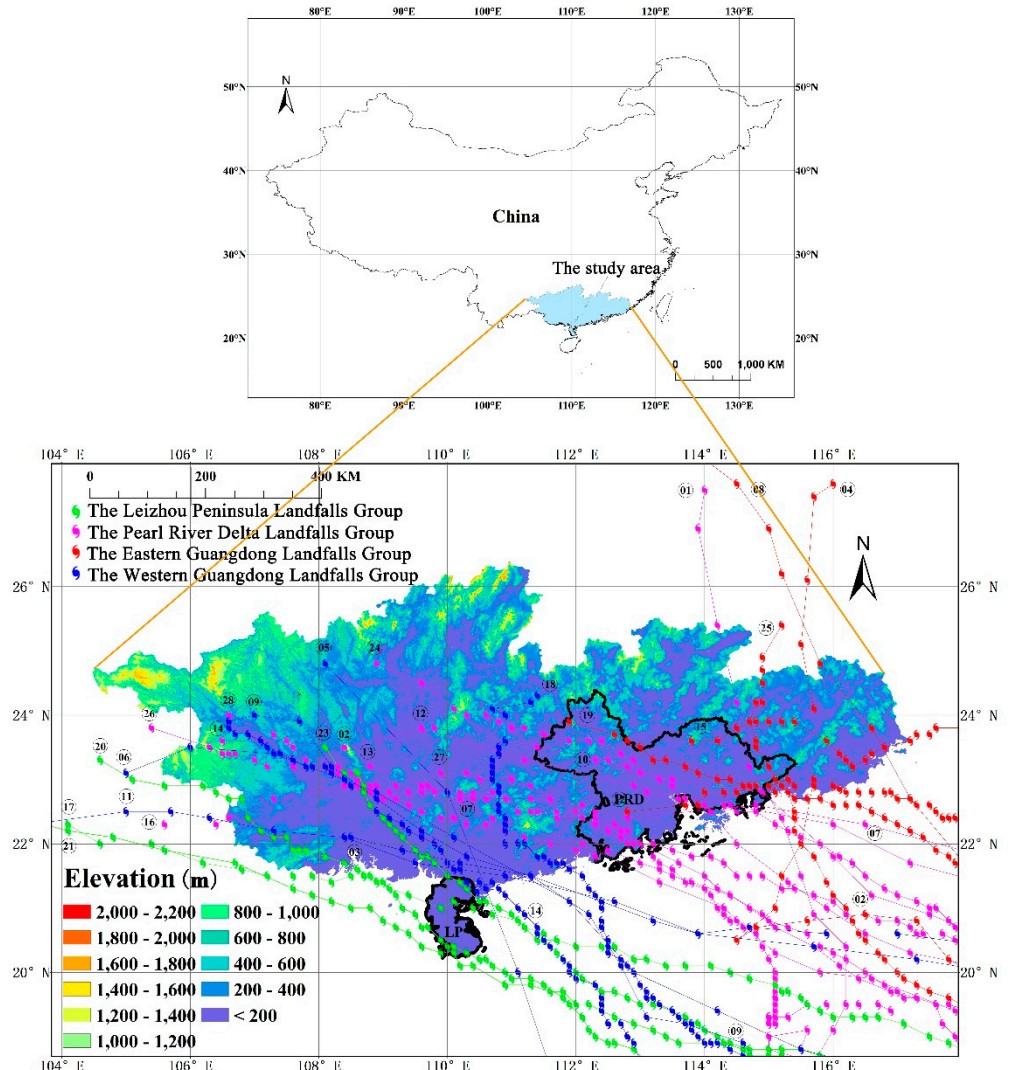

**Figure 2.** The study area and 28 typhoons of four landfall groups in the southeast coastal area of China (SCAC) from 2000 to 2018.

## 3. Methodology

### 3.1. Disaster Vegetation Damage Index (DVDI)

*DVDI* is an index for measuring disaster vegetation damage based on comparing the difference of vegetation condition before and after a disaster [12]. It can be computed by Equation (1).

$$DVDI = mVCI_a - mVCI_b \tag{1}$$

Where $mVCI_a$ and $mVCI_b$ are the $mVCI$ values immediately after and before a disaster, respectively, and $mVCI$ is a modified vegetation condition index which was proposed by Di et al. [12]. The $mVCI$, using median *NDVI* instead of minimum values to calculate vegetation condition index, can avoid cloud contamination of image's pixels and reflects relative vegetation/crop condition comparing with historical records. It can be calculated by Equation (2).

$$mVCI = (NDVI - NDVI_m)/(NDVI_{max} - NDVI_m) \tag{2}$$

Where *NDVI* is the normalized difference vegetation index value of a specific day of interest, $NDVI_{max}$ and $NDVI_m$ are the maximum and median values of *NDVI* respectively for the pixel from the historic records of *NDVI* at the specific day. The $mVCI > 0$ means that the vegetation/crop growth condition is better than historical average (HA) and $mVCI < 0$ means that it is worse than HA [12].

The assumption in *DVDI* is that the vegetation condition (expressed as *mVCI*) should have very small change or no change if the time interval (*b-a*) is small (e.g., a few days) and no vegetative disaster happens during the interval. If vegetation condition get significantly worse during the interval (*DVDI* is a negative value), it indicates a vegetative disaster happens during the interval, and the magnitude of the negative *DVDI* signifies the degree of damage. The positive value of *DVDI* indicates there is no vegetation damage during the time interval, hence, no vegetative disaster. For the typhoon disaster, $DVDI > 0$ means that the observed typhoon has made no damage to vegetation (including forests and crops), and $DVDI < 0$ refers to the degree of damage due to the disaster.

The flowchart for calculation of *DVDI* in GEE is presented in Figure 3. In the image composition step, the two 14-day MODIS image sequences before and after the typhoon are generated, and then $NDVI_a$ and $NDVI_b$ are calculated. In the historic $NDVI_s$ computation step, the historic records of *NDVI* for the corresponding period of the selected typhoon event are created and applied to obtain $NDVI_{max}$ and $NDVI_m$. In the last step, by applying Equations (1) and (2), the *DVDI* value of every pixel in the study area can be calculated and a *DVDI* map can be produced. The key code for deriving *DNVI* in GEE can be accessed at https://code.earthengine.google.com/?accept_repo=users/chuyiwu/calcDVDI.

In order to further identify the degree of damage, the method of equal-interval categorization is used to sort the *DVDI* values into the following five levels: no damage, slight damage, moderate damage, extreme damage, and exceptional damage.

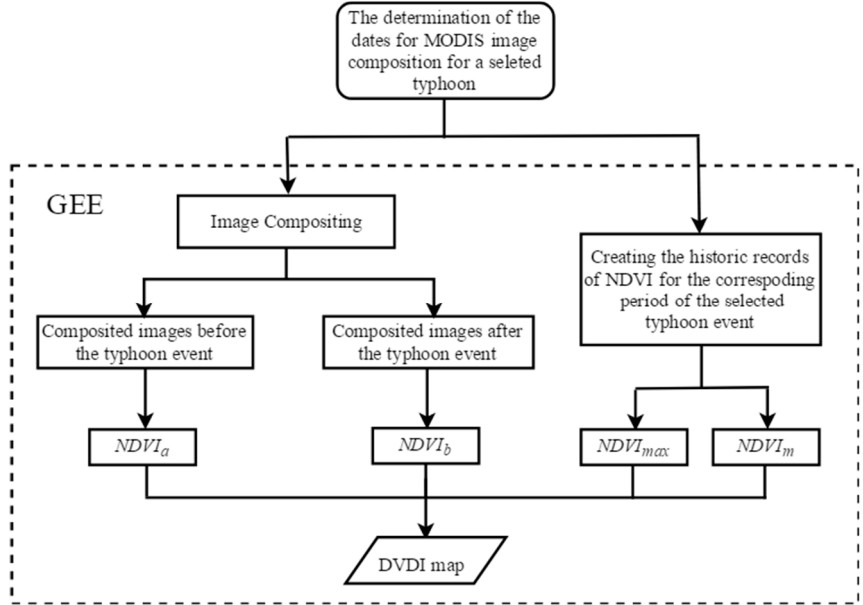

**Figure 3.** The flowchart for calculation of disaster vegetation damage index (*DVDI*) in Google Earth Engine (GEE).

### 3.2. DNDVI and DEVI

*DNDVI* is a most commonly used VD index [9,15,17,18] and can be computed by Equation (3).

$$DNDVI = \frac{NDVI_a - NDVI_b}{NDVI_b} \tag{3}$$

Where $NDVI_a$ and $NDVI_b$ are the *NDVI* values immediately after and before a disaster respectively. Positive *DNDVI* indicates vegetation damage due to the disaster, while negative *DNDVI* represents no VD.

*DEVI* is a frequently used VD index [4,5,18,20] and calculated by Equation (4).

$$DEVI = EVI_a - EVI_b \tag{4}$$

Where $EVI_a$ and $EVI_b$ are *EVI* values immediately after and before a disaster, respectively. *EVI* is calculated from surface reflectance values of red, near infrared, and blue bands (MODIS Bands 1,2 and 3, respectively) by Equation (5).

$$EVI = G \times \frac{NIR - RED}{(L + NIR + C_1 RED + C_2 BLUE)} \tag{5}$$

Where $G$ is the gain factor, $L$ is a parameter of canopy background adjustment, $C_1$ and $C_2$ are coefficients of aerosol resistance term. For MODIS *EVI*, $G = 2.5$, $L = 1$, $C_1 = 6$ and $C_2 = 7.5$ [21].

### 3.3. The Relative Aspect to Typhoon Path

Aspect is defined as the downslope direction of the maximum rate of change in value from each cell to its neighbors [22]. In order to identify the comprehensive effects of aspect and typhoon path on VD, the relative aspect (*RA*) is computed. *RA* acknowledges the relative direction of the aspect to the direction of Typhoon path [7], and is defined as follows: the leeward direction to the typhoon path is

defined as 0°; the windward direction of the typhoon path is defined as 180°; and others are between 0° to 180° [7]. *RA* can be derived by Equations (6) and (7).

$$\text{When } \theta \geq 180, \ RA = \begin{cases} (360 - \theta) + \alpha, \ 0 < \alpha < \theta - 180 \\ \theta - \alpha, \ \theta - 180 \leq \alpha < \theta \\ \alpha - \theta, \ \theta \leq \alpha < 360 \end{cases} \tag{6}$$

$$\text{When } \theta < 180, \ RA = \begin{cases} \theta - \alpha, \ 0 < \alpha < \theta \\ \alpha - \theta, \ \theta - 180 \leq \alpha < \theta \\ (360 - \alpha) + \theta, \ \theta \leq \alpha < 360 \end{cases} \tag{7}$$

Where $\theta$ is the direction of the typhoon path, and $\alpha$ is the real aspect which derived from digital elevation model (DEM).

## 4. Results

### 4.1. The Comparison of DVDI, DEVI and DNDVI Results

Brennan et al. suggested that most forest damage occurred within 100 km of the hurricane path [20]; Zhang et al. proved that the distance from a typhoon's path has a strong influence on VD [7]. It is well known that the distribution of VD is significantly correlated with typhoon path [7,20]. VD maps of DVDI, DEVI and DNDVI of four presentative typhoons (one for a typhoon group) were presented in Figure 4. For Typhoon Mangkhu (Figure 4a–c), comparing the results of DEVI and DVDI, the distribution of DVDI has the strongest correlation with typhoon path, and the distribution of the noise of DVDI is the least; For Typhoons Mujigea (see Figure 4d–f), Vongfong (see Figure 4g–i), Usagi (see Figure 4j–l), and for 18 out of the 24 other typhoons, the same pattern can be found. The wind speed of the typhoon, and hence the vegetation damage, decreases with the distance to the typhoon path. Therefore, the VD represented by the percentage of area suffered from medium or severer vegetative damage should be negatively correlated with distance to the typhoon path. For the study area, 5 km internal buffer is an appropriate unit of measure, since it can reflect DVDI changes and not be too small to induce noise interference. Therefore, for each selected typhoon, the percentages of damaged areas (pixels with DVD/DEVI/DNDVI <0) to the area of SCAC for every 5 km interval buffer of typhoon path were counted. Correlation analysis of the percentages of damaged areas versus buffer distances was executed in SPSS Statistics 20.0. The results are listed in Table 3. Table 3 shows, among DVDI, DEVI, and DNDVI, only DVDI exhibits this negative correlation consistently for all 14 selected typhoons. This proves that DVDI is the best index among DVDI, DEVI, and DNDVI to measure the vegetative damage caused by typhoons. The inconsistent correlations of both DEVI and DNDVI with the distance to typhoon paths (i.e., for some typhoons, the correlation coefficient is negative and for others it is positive) indicate both DEVI and DNDVI are not good indices for measuring the vegetative damage over a large geographic area. Therefore, it can be conclude that DVDI, compared with DEVI and DNDVI, is a more effective index to evaluate VD caused by natural disasters. Below we will focus on using DVDI to explore the spatial characteristics of typhoon-induced VD in the study area.

The values of mVCI, NDVI and EVI of croplands and the other three dominant land cover types may change naturally even without a typhoon during two 14-day periods. In our study, it was supposed the change of vegetation index value by natural growth is much less than that induced by typhoon. Actually, up to 97% of pixels in all 28 typhoons' composite images (see Table 1) are from images acquired within seven days after the typhoon landfall. As shown in the ratio column of Table 1, the percentage of pixels with bad quality (which corresponds to the attribute of data quality as not "0:Highest quality") is rather low even if only using the images acquired within seven days after the typhoon landfall are used to composite the post-typhoon image.

Figure 4 also shows that some areas close to a typhoon's path are identified as slight damage or even no damage. This can be mainly caused by abundant typhoon precipitation causing a possible benefit to vegetation/crop growth in these areas.

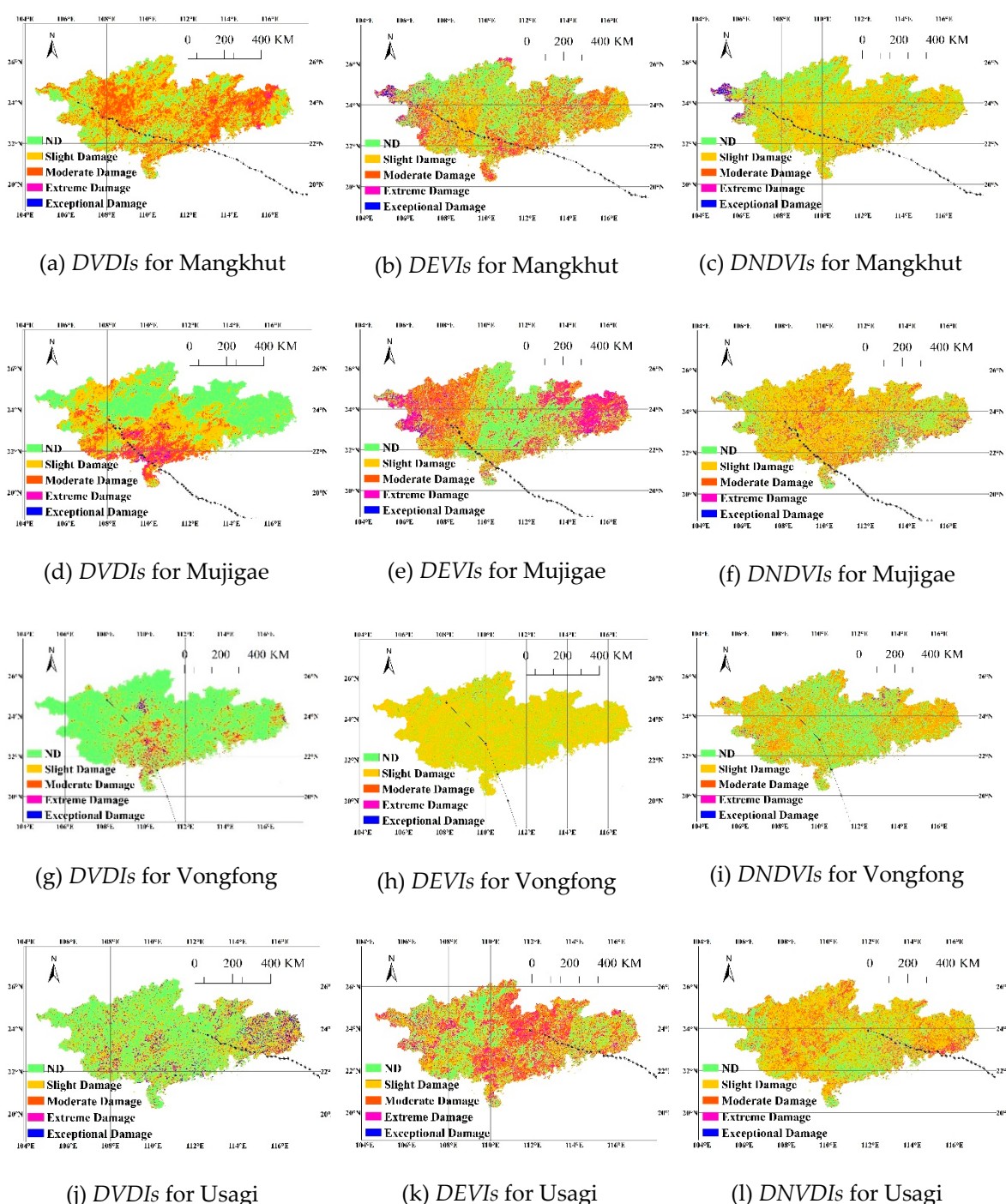

(a) *DVDIs* for Mangkhut

(b) *DEVIs* for Mangkhut

(c) *DNDVIs* for Mangkhut

(d) *DVDIs* for Mujigae

(e) *DEVIs* for Mujigae

(f) *DNDVIs* for Mujigae

(g) *DVDIs* for Vongfong

(h) *DEVIs* for Vongfong

(i) *DNDVIs* for Vongfong

(j) *DVDIs* for Usagi

(k) *DEVIs* for Usagi

(l) *DNVDIs* for Usagi

**Figure 4.** The comparison of *DVDI*, difference in enhanced vegetation index (*DEVI*) and normalized difference vegetation index (*DNDVI*) results fortyphoon Mangkhut (201822), Mujigae (201522), Vongfong (200214) and Usagi (201319): (**a**,**d**,**g**,**j**) are DVDIs for the 4 presentative typhoons respectively; (**b**,**e**,**h**,**k**) are DVDIs for the 4 presentative typhoons respectively; (**c**,**f**,**i**,**l**) DVDIs for the 4 presentative typhoons respectively.

**Table 3.** Correlation analysis results of DVDI/DNDV/DEVI and area percentages of VD.

| Typhoon | Group | DVDI | | DNDVI | | DEVI | |
|---|---|---|---|---|---|---|---|
| | | R | SIG | R | SIG | R | SIG |
| Vongfong | Group 4 | −0.883 | 0.000 | 0.773 | 0.000 | 0.251 | 0.036 |
| Mugigea | Group 1 | −0.971 | 0.000 | −0.579 | 0.000 | −0.606 | 0.000 |
| Usagi | Group 3 | −0.544 | 0.000 | 0.010 | 0.932 | −0.628 | 0.000 |
| Mangkhut | Group 3 | −0.091 | 0.455 | −0.627 | 0.000 | −0.670 | 0.000 |
| Utor | Group 2 | −0.882 | 0.000 | −0.676 | 0.000 | −0.682 | 0.000 |
| Dujuan | Group 2 | −0.865 | 0.000 | −0.748 | 0.000 | −0.525 | 0.000 |
| Nuri | Group 2 | −0.232 | 0.054 | −0.412 | 0.000 | −0.170 | 0.159 |
| Kai-tak | Group 1 | −0.298 | 0.012 | −0.558 | 0.000 | 0.030 | 0.804 |
| Rammasun | Group 1 | −0.787 | 0.000 | 0.182 | 0.131 | −0.588 | 0.000 |
| Kalmaegi | Group 1 | −0.862 | 0.000 | 0.747 | 0.000 | 0.961 | 0.000 |
| Sanvu | Group 3 | −0.809 | 0.000 | −0.637 | 0.000 | 0.342 | 0.004 |
| Prapiroon | Group 4 | −0.212 | 0.078 | 0.456 | 0.000 | −0.177 | 0.142 |
| Imbudo | Group 4 | −0.370 | 0.002 | 0.569 | 0.000 | 0.956 | 0.000 |
| Utor | Group 4 | −0.864 | 0.000 | 0.321 | 0.007 | 0.285 | 0.017 |

*4.2. The Spatial Characteristics of Historical Landfalling Typhoons in SCAC*

### 4.2.1. The Accumulated DVDIs of 28 Selected Typhoons

The 28 *DVDI* images for the selected typhoons were calculated and downloaded from GEE and overlaid in ArcGIS 10.2. The statistics of 28 typhoons' DVDI maps shows that, when DVDI <10, the highest area percentage for one typhoon is 3.32%, and the area percentages of 18 typhoons are less than 1%. Therefore, in order to calculate the magnitude of VD, we set *DVDI* to 0 when *DVDI* >0, and set *DVDI* to −10 when *DVDI* < −10. The spatial distribution of the accumulated *DVDIs* of 28 selected typhoons is presented in Figure 5. VD is more severe in the southern and central regions than in the north, due to two reasons: the northern region is further from the coastline and has higher elevation. Figure 5 also shows that VD in the west of Pearl River Delta is most severe (in red ellipse), while the elevation of this severe VD region is almost lower than 600 m. It includes most parts of Zhaoqing, Yunfu, Yingde, Wuzhou cities, and also some parts of Jiangmen, Heyuan, Huizhou, Guangzhou, Hezhou, Yulin, and Guigang cities.

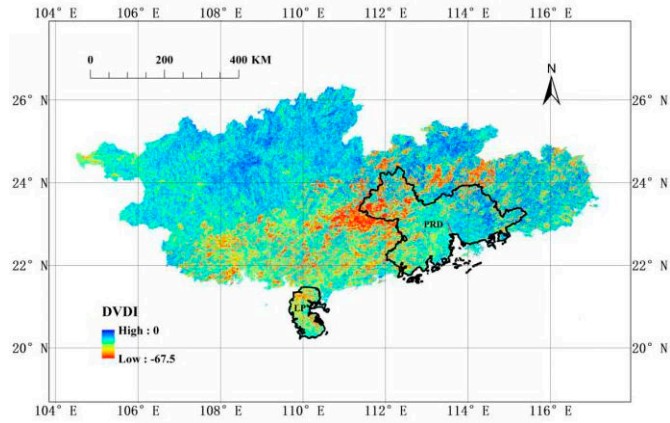

**Figure 5.** The spatial distribution of accumulated *DVDIs* of 28 selected typhoons from 2000 to 2018.

### 4.2.2. The accumulated DVDIs of Four Typhoon Groups

Based on the rule of mapping of Figure 5 (i.e., set *DVDI* to 0 when *DVDI* >0, and set *DVDI* to −10 when *DVDI* < −10), the accumulated *DVDI* maps of four typhoon groups are also delineated respectively (see Figure 6). Figure 6 shows that severe VD regions with the lowest accumulated *DVDIs*

of four typhoon groups are evidently spatially correlated with the typhoons' landfall locations: for the Pearl River Delta landfall group (Figure 6a), the severe VD region (in red ellipse) is around the Pearl River Delta; for the Western Guangdong landfall group (Figure 6b), the severe VD region is in the west of Pearl River Delta; for the Eastern Guangdong landfall group (Figure 6c), the northern and western regions of Pearl River Delta are both severe VD regions; and for the Leizhou Peninsula landfalls group (Figure 6d), the severe VD region is around the Leizhou Peninsula.

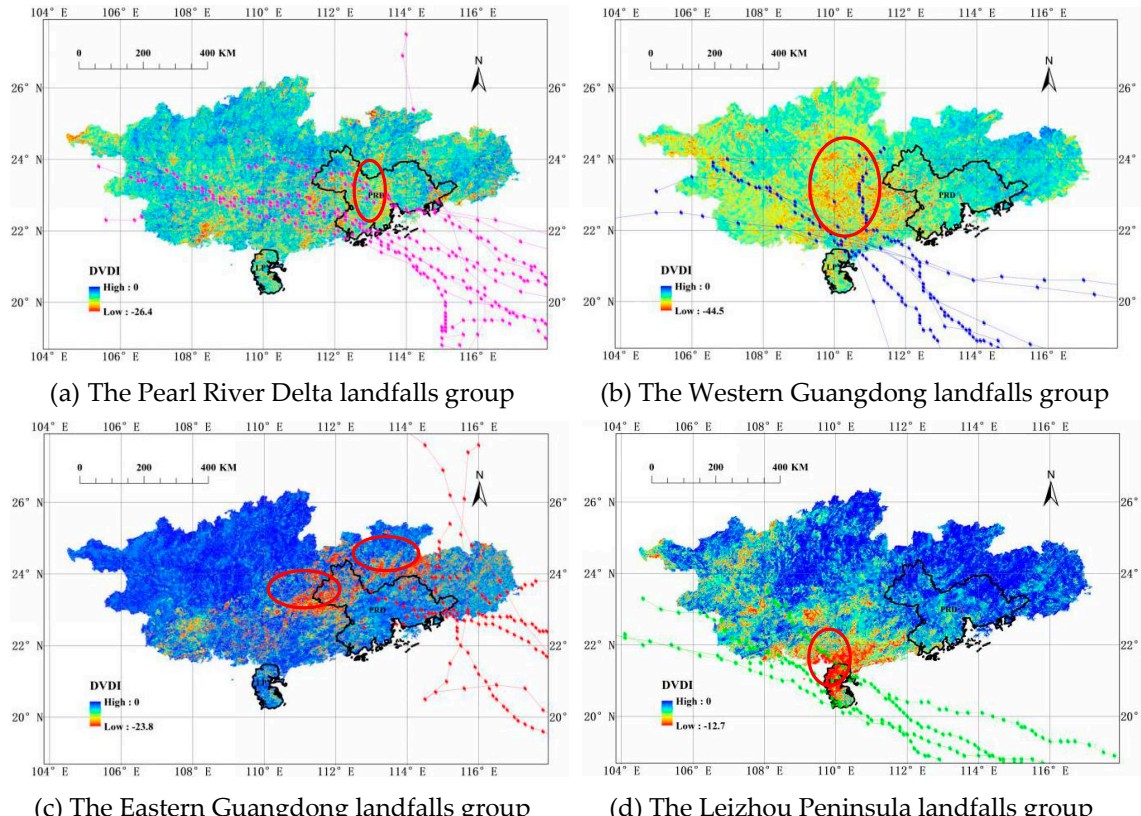

(a) The Pearl River Delta landfalls group

(b) The Western Guangdong landfalls group

(c) The Eastern Guangdong landfalls group

(d) The Leizhou Peninsula landfalls group

**Figure 6.** The spatial distributions of accumulated *DVDIs* of four groups: (**a**) The Pearl River Delta landfalls group; (**b**) The Western Guangdong landfalls group; (**c**) The Eastern Guangdong landfalls group; (**d**) The Leizhou Peninsula landfalls group.

### 4.2.3. Vegetation Damages Over Different Land Covers

Percentages of damaged areas (DVDI <0) over the four dominant land cover types were computed and shown in solid lines (in Figure 7). Figure 7 illustrated that forest, with the largest area among eight land cover types (see Table 2), has the greatest area of damage, followed by sparse forests, croplands, and impervious lands. The similitude of line shapes of forests, sparse forest and croplands shows that vegetation damages over these three land cover types are similar, while impervious lands' line keeps stable. This possibly means its vegetation damage changes less, or its VDs' fluctuation cannot be discovered in items of its smallest area. Considering the total area of each land cover type, the percentage of damaged area for the type was recalculated and shown by dashed lines. Compared with solid lines, dashed lines can enlarge and better present VD changes, and show very similar shapes.

The possible reasons for little VDs of different land cover types are as follows: forests, including evergreen needleleaf and broadleaf forests, deciduous broadleaf forests, and mixed forests, are in the best growth period from July to October in our study time period, and, relatively invulnerable to strong wind and heavy rain, are also relatively hard to recover. Sparse forests are usually mixed woody savannas with natural vegetation, croplands mixed crops with natural vegetations, and impervious lands mix urban and built-up land with planted grasses and trees. Comparing with forests, sparse

forests and croplands are more easily damaged, but by means of natural vegetation growing during the short period of typhoon landing and extinction, their VDs situation can be improved to some extent. Though planted grasses in impervious lands are most invulnerably damaged, planted trees (usually separated) are sensitive to strong wind.

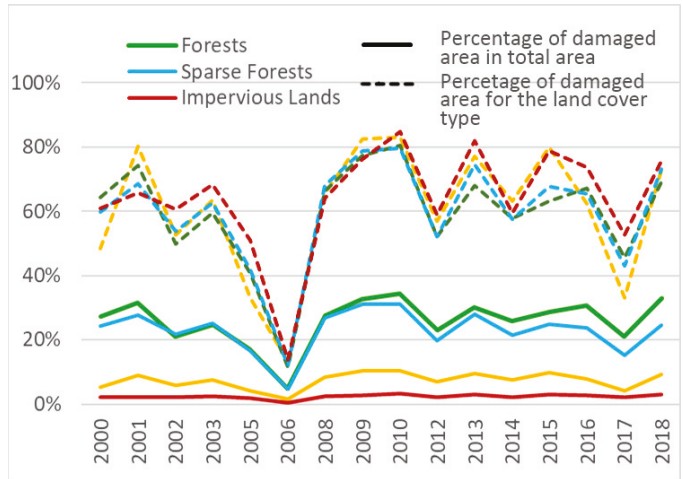

**Figure 7.** The percentages of damaged areas over the four dominant land cover types and the percentages of damaged area for the type in SCAC from 2000 to 2018.

### 4.2.4. Influence of Topography on Vegetation Damages by Typhoons

For each selected typhoon, the percentages of damaged areas (pixels with DVDI <0) to the area of SCAC for every 200 m elevation level were counted. Correlation analysis of the percentages of damaged areas versus elevations was executed in SPSS. The results are listed in Table 4.

**Table 4.** The area percentages of 11 elevation levels.

| Elevation Level | Area Percentage | Elevation Level | Area Percentage |
|---|---|---|---|
| <200 | 47.78% | 1200–1400 | 0.94% |
| 200–400 | 22.65% | 1400–1600 | 0.37% |
| 400–600 | 13.16% | 1600–1800 | 0.10% |
| 600–800 | 7.73% | 1800–2000 | 0.01% |
| 800–1000 | 4.97% | 2000–2200 | 0.00% |
| 1000–1200 | 2.29% | | |

Table 4 shows that, among 28 selected typhoons, there are 13 typhoons with negative correlation coefficient (elevation increases, damaged areas decreases), and 15 typhoons with positive correlation coefficient (elevation increases, damaged areas also increases). As at 0.01 significance level, 10 and 11 typhoons obtained negative and positive correlation between the percentages of damaged areas and elevations respectively. Therefore, the conclusion of no influence of topography on vegetation damages by a single typhoon event in SCAC can be made. One explanation could be that the elevation in 96.3% of the study area is below 1000 meters above sea level and the strong circular wind of a typhoon extends from near ground to several thousand meters above sea level.

### 4.2.5. Influence of Relative Aspect on Vegetation Damages by Typhoons

Typhoons Mujigea (201522, Group 1), Mangkhut (201822, Group 2), Usagi (201319, Group 3), and Vongfong (200214, Group 4) were selected for analyzing the influence of relative aspect on VD. Vegetation damages induced by the four representative typhoons under different relative aspects are presented in Figure 8. The percentages of decreased DVDI curves show the accumulated percentages

of damaged areas of four levels (slight, moderate, extreme, and exceptional damage). For Typhoons Mujigea and Mangkhut, the percentages of damaged areas in the windward direction are slightly higher than those in the leeward direction, and the differences in the percentages of damage for every 20° RA are less than 3%, while for Typhoons Usagi and Vongfong, the percentages of damaged areas keep stable when RAs changes. Thus the correlation between *DVDIs* and *RAs* is insignificant. The possible reason of the insignificance of *DVDIs* and *RAs* is that instantaneous wind direction can be different from the typhoon paths and changes frequently.

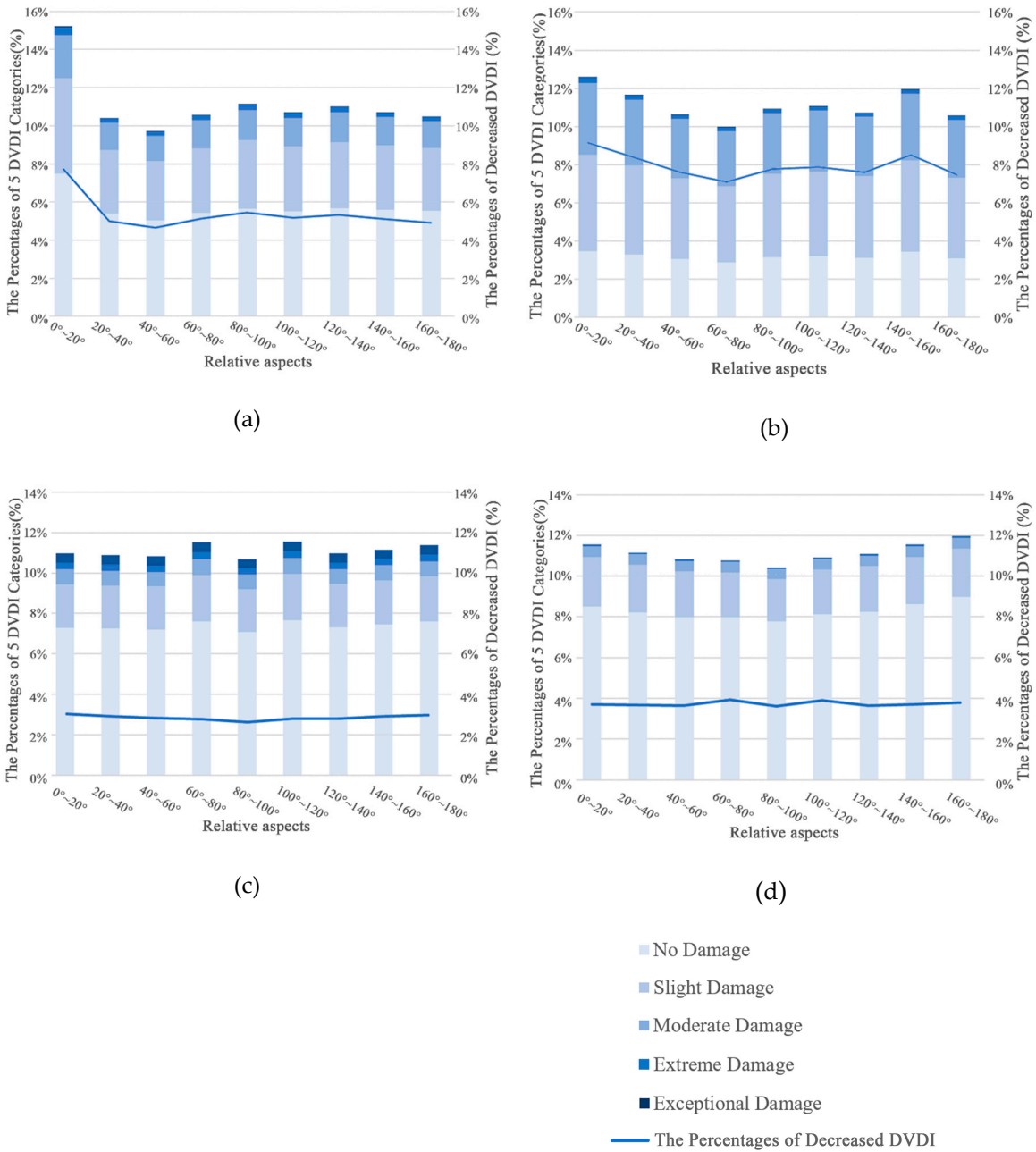

**Figure 8.** The percentages of damaged areas for different relative aspects. (**a**) Typhoon Mujigae (201522), (**b**) Typhoon Mangkhut (201822), (**c**) Typhoon Usagi (201319), (**d**) Typhoon Vongfong (200214).

## 5. Discussion

In order to validate the performance of DVDI, we include the comparison of the damaged cropland area calculated from DVDI maps for areas with moderate or above levels of damage, as obtained

from the Yearbook of Meteorological Disasters in China (2005–2018) for the 20 typhoons listed in the Yearbook. Figure 9 is the plot of damaged areas for the 20 typhoons. From the figure, it can be found that damaged area calculated from DVDI follows the general trend of damaged cropland area reported by the yearbook, particularly well for the Guangdong province (orange lines) but not so well for Guangxi province (blue lines). One explanation for this could be that the damaged hectares reported by the yearbook are the cropland which actually lost yield, while DVDI measures the change of spectral signal which could be caused by loss or damage of leaves and some crops can recover from such damage later. Such a leaf damage can be caused by winds that moderately strong but not strong enough to destroy the entire crop. This could also explain why the agreement of damaged areas in Guangxi province is poorer than Guangdong province because Guangxi province is more inland than Guangdong province and typhoons reaching to Guangxi generally have much weaker winds, which could damage the crop leaves but may not devastate the crop and yield. Another phenomenon can be found from the figure is that the area damaged by typhoons estimated by DVDI is generally larger than reported in the yearbook. This could be caused by defining the threshold values for moderate and severer damage classes too high. Further research on the class boundaries for defining damage classes in DVDI is needed. It is also interesting to find that, for both the Kai-Tak and Hagupit typhoons, the DVDI and YMDC seem to have a close match for both Guangxi and Guangdong provinces. A further study should be conducted to identify the specific characteristics of those two typhoons which resulted in the close match. Based on above observations, it can be concluded DVDI is a good index to measure vegetative damage due to typhoons, and the damaged cropland area calculated from DVDI could be used as a quick estimation of the area of crop loss, especially for the geographic area near the typhoon landfall.

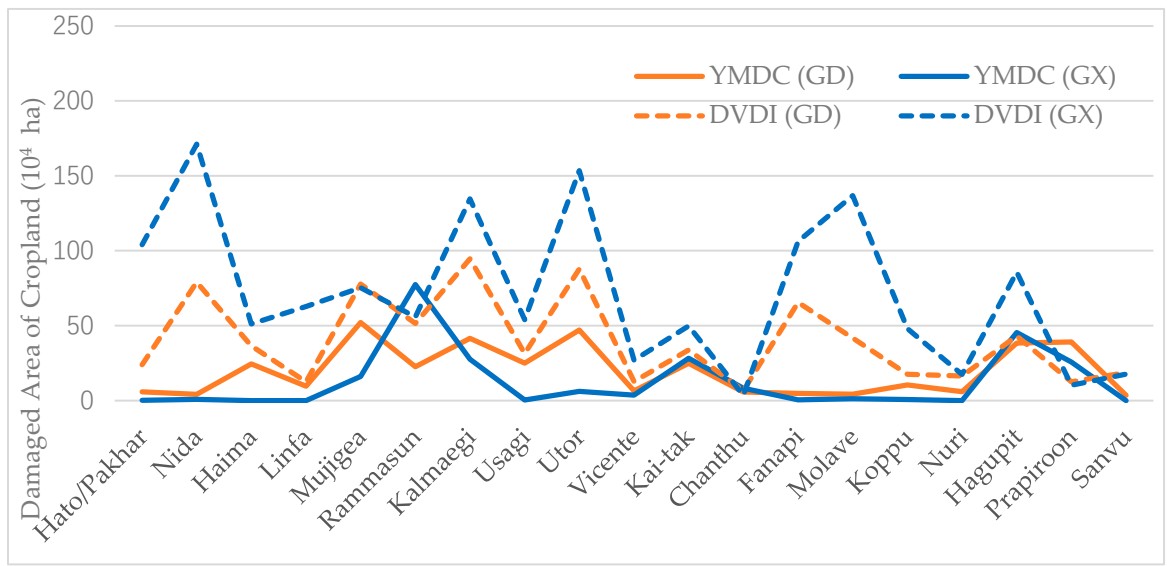

**Figure 9.** The comparison of the damaged cropland area calculated from DVDI maps for area with moderate or above levels of damage (DVDI, dashed lines) and that obtained from the Yearbook of Meteorological Disasters in China (YMDC, solid lines) (2005–2018). Orange and blue lines are for Guangdong and Guangxi Provinces respectively.

## 6. Conclusions

Using MODIS data in GEE, DVDI, DEVI, and DNDVI values for the 28 selected typhoons landing in SCAC were calculated and compared. The DVDI images were overlaid with land cover, elevation, relative aspect and typhoon path layers, and the spatial characteristics of vegetation damages induced by historical typhoons were explored with the aid of spatial statistical analysis in ArcGIS. The results showed that:

1.  DVDI is a more effective index for evaluating VD caused by typhoons.
2.  The Pearl River Delta, with elevation almost less than 600 m, is the most severe VD region. The severe VD regions for four typhoon groups have significant spatial correlation with landing locations.
3.  Forests are ranked first in terms of damaged areas by typhoon in every year, followed by sparse forests, while the percentage of damaged area for the land cover type is very similar.
4.  Topography has no influence on VD by a single typhoon event, and RA has no correlation with VD per typhoon in SCAC.

**Author Contributions:** Conceptualization, L.L. and L.D.; Methodology, L.L. and C.W.; Software, C.W.; Validation, L.L.; Formal Analysis, L.L. and C.W.; Investigation, C.W.; Resources, C.W.; Data Curation, C.W.; Writing-Original Draft Preparation, L.L.; Writing-Review and Editing, L.L and L.D.; Visualization, C.W.; Supervision, L.L. and L.D.; Project Administration, L.L.; Funding Acquisition, L.L. All authors have read and agreed to the published version of the manuscript.

**Funding:** This research is supported in part by a grant from China's National Science and Technology Support Program grant (National Key R&D Program of China, 2018YFB0505000).

**Acknowledgments:** We would like to thank four anonymous reviewers for the comments and suggestions that significantly help to improve the quality of the paper, and thank editor Milica Kovacevic for the suggestions and reminders. We would also thanks Yue Zhu, a graduate student from Zheijiang University for the data preprocessing.

**Conflicts of Interest:** The authors declare no conflict of interest.

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
