# Peer review of "Exploring the Spatial Characteristics of Typhoon-Induced Vegetation Damages in the Southeast Coastal Area of China from 2000 to 2018"

_remotesensing, doi:10.3390/rs12101692_

Round 1

Reviewer 1 Report

The authors have successfully addressed reviewers' comments. Overall, the quality of the manuscript is significantly improved.

Author Response

Dear reviewer, 

We greatly appreciate your valuble comments. 

Best regards!

Reviewer 2 Report

The manuscript has a lot of issues with its structure as in the Results and Discussion section the authors refer to methodological steps which are not mentioned at all in the methodology section leading to confusion and difficulty to the reader to follow and understand their methodological framework. If those are moved to the Methodology which lacks of explanatory details as it stands, it will strengthen that section and improve the understanding of their overall methodological framework idea and outcomes.

See attachment for further in detail comments.

Author Response

Dear Reviewer:

We greatly appreciate your valuable comments. We have revised our manuscript based on your comments and uploaded the revised manuscript to Remote Sensing website. We hope the revised paper meets your expectation.Our responses to your comments are listed in the attached. If you have any questions, please don’t hesitate to contact us.

Reviewer 3 Report

  1. The validation discussion part (page 12-13), the area damaged by typhoons estimated by DVDI is generally larger than reported in the Yearbook. The authors think "This could be caused by defining the threshold values for moderate and severer damage classes too high."  Another potential reason is that the authors didn't consider vegetation phenology, the natural seasonal change of vegetation was counted as the impacts of Typhoon.  Please discuss the impacts of natural change of vegetation
  2. The validation discussion part (page 12-13). The result for Guangdong Province is much better than Guangxi Province.  The main reason is the intensity of Typhoon. If have wind information, can do statistical analysis of  DVDI changes with wind speed. If don't have wind data, can compare DVDI changes of areas along/close to Typhoon path and areas far away from Typhoon path to see if there are signifcant differences. 

Author Response

Dear Reviewer:

We greatly appreciate your valuable comments. We have revised our manuscript based on your comments and uploaded the revised manuscript to Remote Sensing website. We hope the revised paper meets your expectation. Our responses to your comments are listed in attached. If you have any questions, please don’t hesitate to contact us.

Round 2

Reviewer 2 Report

Comments to be acknowledged by authors:

Line 73: Correct “deamage” to “damage”

Line 86: Correct “downloadin” to “downloading”

Line 204: Section 3.4 consists of repetition. Move section 3.4 and merge it with the one of 3.1, where DVDI is explained to avoid repetition. Authors can just simply move flowchart in Section 3.1 and refer to the flowchart regarding the steps followed in GEE or just move the section below the 3.1 as 3.2. Then authors can also refer to check for further details the accompanied GEE script code, which is not added throughout the manuscript and needs to be presented to the supplementary material of this manuscript, as this seems to be the highlight and the major contribution of the manuscript to the scientific society.

Line 229: Correct “5 KM” to “5 km”, and just briefly refer to why 5km internal buffer was selected. Why not more or less?

Line 290 to 292: Change to: “…croplands shows that vegetation damages over these three land cover types are similar, while impervious lands’ line keeps stable. This possibly means its vegetation damages changes less, or its VDs’ fluctuation cannot be discovered in items of its smallest area.”

Authors are using quite often within the manuscript the personal pronoun “we” and should be avoided in any academic or formal writing. Please modify accordingly as the language should therefore be impersonal within the manuscript.

Author Response

Dear Reviewer: 
We greatly appreciate your valuable comments. We have revised our manuscript based on your comments and uploaded the revised manuscript to Remote Sensing website, please see the attached. We hope the revised paper meets your expectation. If you have any questions, please don’t hesitate to contact us.

Best regards!
